# End-to-End Edge-Guided Multi-Scale Matching Network for Optical Satellite Stereo Image Pairs

**Yixin Luo** [1,2,3], **Hao Wang** [1,2,3] and **Xiaolei Lv** [1,2,3,*]

1   Key Laboratory of Technology in Geo-Spatial Information Processing and Application System, Aerospace Information Research Institute, Chinese Academy of Sciences, Beijing 100190, China; luoyixin21@mails.ucas.ac.cn (Y.L.); wanghao197@mails.ucas.ac.cn (H.W.)
2   Aerospace Information Research Institute, Chinese Academy of Sciences, Beijing 100094, China
3   School of Electronic, Electrical and Communication Engineering, University of Chinese Academy of Sciences, Beijing 100049, China
*   Correspondence: academism2017@sina.com

**Abstract:** Acquiring disparity maps by dense stereo matching is one of the most important methods for producing digital surface models. However, the characteristics of optical satellite imagery, including significant occlusions and long baselines, increase the challenges of dense matching. In this study, we propose an end-to-end edge-guided multi-scale matching network (EGMS-Net) tailored for optical satellite stereo image pairs. Using small convolutional filters and residual blocks, the EGMS-Net captures rich high-frequency signals during the initial feature extraction phase. Subsequently, pyramid features are derived through efficient down-sampling and consolidated into cost volumes. To regularize these cost volumes, we design a top–down multi-scale fusion network that integrates an attention mechanism. Finally, we innovate the use of trainable guided filter layers in disparity refinement to improve edge detail recovery. The network is trained and evaluated using the Urban Semantic 3D and WHU-Stereo datasets, with subsequent analysis of the disparity maps. The results show that the EGMS-Net provides superior results, achieving endpoint errors of 1.515 and 2.459 pixels, respectively. In challenging scenarios, particularly in regions with textureless surfaces and dense buildings, our network consistently delivers satisfactory matching performance. In addition, EGMS-Net reduces training time and increases network efficiency, improving overall results.

**Keywords:** dense matching; end-to-end stereo matching network; pyramid feature; multi-scale integration; trainable guided filter

## 1. Introduction

The digital surface model (DSM) serves as the cornerstone for three-dimensional (3D) reconstruction, capturing authentic ground undulations and finding applications in various fields, including change monitoring and urban planning [1,2]. The stereo matching of remote sensing satellite image pairs plays a pivotal role in the DSM production process [3]. Compared to the direct acquisition of DSM by light detection and ranging [4], dense stereo matching methods offer advantages such as lower cost and higher automation, making them widely applicable for recovering 3D information from imagery. Using two remote sensing images taken by the same camera from different viewpoints, epipolar rectification is used to create left and right stereo images [5]. This process ensures that each pixel and its corresponding pixel are aligned on the same row in both images. The disparity is then calculated as the difference between the column numbers of the corresponding pixels. The goal of stereo matching is to produce a disparity map that is converted to a depth map using geometric relationships to recover elevation information. The accuracy of the disparity map derived from stereo matching significantly affects the accuracy of the resulting DSM. Therefore, improving the accuracy of stereo matching technology has become a major research focus.

Through years of dedicated research, traditional stereo matching technology has evolved from local and global matching [6,7] to semi-global matching [8,9]. This evolution has improved matching accuracy and achieved breakthroughs in matching speed. However, traditional methods suffer from inherent shortcomings such as a large number of parameters and insufficient processing power for complex scenes [10]. These limitations are particularly apparent in optical satellite imagery, where traditional methods struggle due to the long baseline and large coverage area. In recent years, stereo matching using deep learning technology has made steady progress and achieved remarkable results. Early approaches in this field replace certain steps within traditional methods with deep learning techniques [11–13]. A notable example is [14], which uses the convolutional neural networks (CNNs) with shared weights for feature extraction and cosine similarity to compute the probability of matching between two image blocks. While these methods have made remarkable progress over traditional approaches, they still face inherent challenges in pathological regions, such as textureless areas, occluded regions and repetitive patterns. Therefore, there is an urgent need to develop end-to-end stereo matching networks that use deep learning techniques at all stages. Such approaches can seamlessly integrate global information into the network to optimize matching results.

The process of end-to-end stereo matching networks can be divided into several modules, broadly categorized as 3D cost volume, four-dimensional (4D) cost volume and hybrid 3D-4D cost volume, based on different methods of cost volume construction [10]. Some networks [15,16] use conditional random fields to build a recurrent neural network for 3D cost volume regularization. Ref. [17] is an early adopter, introducing appropriate operators to construct the 3D cost volume and inspiring subsequent architectures [18–20]. In [21], the network uses input images to construct residuals to optimize initial disparity estimation, demonstrating remarkable performance in both training results and speed of operation.

While the 3D cost volume is highly efficient, it suffers from the elimination of the feature dimension, resulting in a final performance that is not as robust as that achieved with the 4D cost volume. In [22], disparity features are represented by 4D cost volumes and consolidated using 3D CNNs. A notable breakthrough in disparity estimation is introduced by incorporating soft argmin into the regression process. Building upon these foundations, many end-to-end models have been introduced. For example, pyramid stereo matching network (PSMNet) [23] integrates spatial pyramid pooling layers into the network to enhance its ability to exploit contextual information. It also uses stacked multiple hourglass networks for cost volume regularization. StereoNet [24] is recognized as the first real-time stereo matching network. It performs initial disparity estimation in low-resolution cost volumes and introduces a reference image in the final step to generate residuals. This process ensures that the up-sampling phase recovers high-frequency detail and produces an edge-preserved disparity map. While these methods produce excellent results, the extensive use of 3D CNN parameters for aggregation increases computational complexity and lengthens training time. To address this challenge, several networks have introduced innovations [25]. For example, a practical deep stereo network [26] compresses the features of the associated left and right images before cost aggregation to reduce memory requirements. Previous studies detailing these innovations are summarized in Table 1.

In summary, existing stereo matching methods are not tailored for optical satellite imagery. Complex networks have high hardware requirements, making them unsuitable for processing large areas of optical imagery. On the other hand, simple networks struggle to handle the rich content and wide disparity range of optical imagery, presenting additional challenges. Traditional approaches, such as cropping images into smaller blocks to reduce memory consumption, can compromise image integrity and affect training results. Therefore, a balance must be struck between minimizing training time and maintaining effectiveness when handling optical satellite imagery.

To address the challenge of dense matching in large and complex scenarios of optical satellite imagery, this paper introduces an end-to-end edge-guided multi-scale matching

network, guided by the insights provided in Table 1. The main contributions of this paper are as follows:

1.  In feature extraction, we first use cascaded small convolutional filters and residual blocks to learn features at the original resolution. This ensures that the high-frequency information in the image is fully captured. Subsequently, the size is reduced to one-fourth of the original image to reduce computational complexity.
2.  We use an efficient down-sampling operation to extract pyramid features at different scales, simultaneously increasing the number of channels while decreasing the resolution. This approach minimizes the loss of information during the down-sampling process and better preserves the extracted feature information.
3.  We construct 4D cost volumes based on features extracted at different scales and design a top–down cost volume fusion module. Within this module, we incorporate the Squeeze-and-Excitation (SE) block, which recalibrates channels based on feature importance to provide more accurate feature information through multi-scale fusion.
4.  We introduce a disparity refinement module where the left image is trained to generate a guidance map. This guidance map is then fed into a trainable guided filtering layer along with a low-resolution disparity map. The result is a refinement process that enhances finer edges when up-sampling from the low-resolution disparity map to the original resolution.

**Table 1.** The summary of the previous work and the proposed solution.

| Related Work | Weaknesses | Proposed Solution |
| --- | --- | --- |
| [11–14] | Lack of global information | Create an end-to-end stereo matching network |
| [15–21] | 3D cost volume eliminates feature dimensions | Use 4D cost volume |
| [22,23] | Large memory usage and long training time | Optimize network structure and improve information access |
| [24] | Fast but ineffective | Set up a multi-scale feature extraction network |
| [25–27] | Unclear edges | Add disparity refinement module and optimize images using reference maps |

This paper is organized as follows: Section 2 introduces the specific architecture of the proposed network. Section 3 provides a detailed description of the experimental materials. Results and analysis are presented in Section 4. Finally, discussions and conclusions are given in Section 5.

## 2. Methods

Considering the characteristics of high-resolution optical satellite stereo image pairs, we opt for the 4D cost volume to ensure the accuracy of the results and propose an edge-guided multi-scale matching network. The whole network architecture consists of five modules: feature extraction, cost volume construction, cost volume aggregation, disparity regression and disparity refinement. The structure of the EGMS-Net is shown in Figure 1. In the first step, multi-scale pyramid features are extracted from the input image pair using a feature extraction network with shared weights and efficient down-sampling networks. Next, 4D cost volumes are constructed separately at multiple scales, and a multi-scale cost volume fusion network aggregates the cost volumes from top to bottom using a 3D CNN. In the disparity regression, soft argmin is used to estimate the disparity. To optimize the low-resolution disparity maps generated due to memory and speed limitations, we use fast trainable guided filters and introduce the left image as a reference image to restore the original resolution disparity map. The details and specific implementation processes for each part are presented in the following subsections.

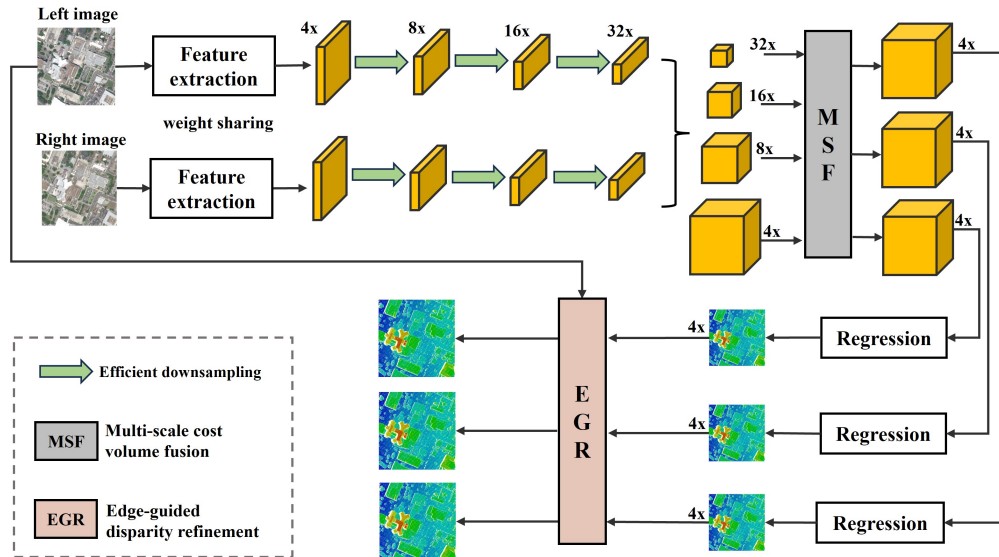

**Figure 1.** The schematic structure of the EGMS-Net.

## 2.1. Feature Extraction with Efficient Down-Sampling

Feature extraction serves as the initial phase in the network, which is critical for retrieving localised information from each pixel (image block) to ensure accurate matching. Many networks choose to reduce the resolution initially to reduce the computational cost. For example, PSMNet and StereoNet use convolutional kernels with a stride of 2 to reduce the resolution of the feature map by half and 1/8 of the original resolution, respectively. These networks are tailored for ground scenes with many textureless regions, where resolution reduction is necessary to ensure feature extraction from a large receptive field. However, unlike ground scenes, optical satellite imagery contains a wide variety of features, including small and dense buildings. Reducing resolution at the outset risks losing fine detail. Therefore, we use three small convolutions for feature extraction at the original size to minimise the loss of information due to resolution reduction.

We also increase the number of channels from 32 to 64, a common operation in networks [23,24,27]. This increase helps to extract richer feature representations, fully capturing high-frequency signals and increasing the expressiveness of the model. Once sufficient high-frequency information is obtained, two residual blocks [28] with a stride of 2 are used to reduce the resolution of the feature map to 1/4 of the original, allowing for faster training. Multiple residual blocks are then used to further extract features at lower resolutions and increase network depth. This results in a larger receptive field, allowing more features to be captured from similar pixels, especially in textureless regions. The feature extraction architecture is outlined in Table 2. In this part, we use a Siamese network to share weights, allowing both input images to generate their own feature maps.

Satellite images contain a wide variety of objects of different sizes. To capture the detail and hold more information, we construct pyramid features with different scales and receptive fields [27]. The low-resolution feature maps are tailored to resolve the interference of textureless areas while maintaining a compact arrangement of feature vectors. Conversely, the high-resolution feature maps are designed to recover fine local features of objects.

**Table 2.** The architecture of the feature extraction.

| Layer | Setting | Output |
|---|---|---|
| Input | / | $H \times W \times 1$ |
| conv0_1 | $3 \times 3 \times 16$ | $H \times W \times 16$ |
| conv0_2 | $3 \times 3 \times 32$, dila = 2 | $H \times W \times 32$ |
| conv0_3 | $3 \times 3 \times 64$, dila = 4 | $H \times W \times 64$ |
| conv1_1 | $\begin{bmatrix} 3 \times 3 \times 64 \\ 3 \times 3 \times 64 \end{bmatrix} \times 5$, stride = 2 | $H/2 \times W/2 \times 64$ |
| conv1_2 | $\begin{bmatrix} 3 \times 3 \times 64 \\ 3 \times 3 \times 64 \end{bmatrix} \times 5$, stride = 2 | $H/4 \times W/4 \times 64$ |
| conv1_3 | $\begin{bmatrix} 3 \times 3 \times 64 \\ 3 \times 3 \times 64 \end{bmatrix} \times 3$ | $H/4 \times W/4 \times 64$ |

To capture features of different scales, many stereo matching networks use pooling layers to reduce the size of feature maps. The principle outlined in [29] emphasizes that, as the network deepens, the size of the feature maps should gradually decrease to avoid excessive feature reduction, which can lead to information loss. To address this, it is advisable to increase the number of channels before merging layers to avoid rendering bottlenecks. In our approach, we use a more efficient down-sampling scheme, as shown in Figure 2. We use $(1 \times 1)$ kernels to compress feature channels, which increases network complexity without compromising accuracy. We also replace large filters $(5 \times 5)$ with two small convolutional filters $(3 \times 3)$ and concatenate features from different receptive fields. This reduces computational complexity and speeds up training. The final output feature map sizes are 1/4, 1/8, 1/16 and 1/32 of the input image size, respectively.

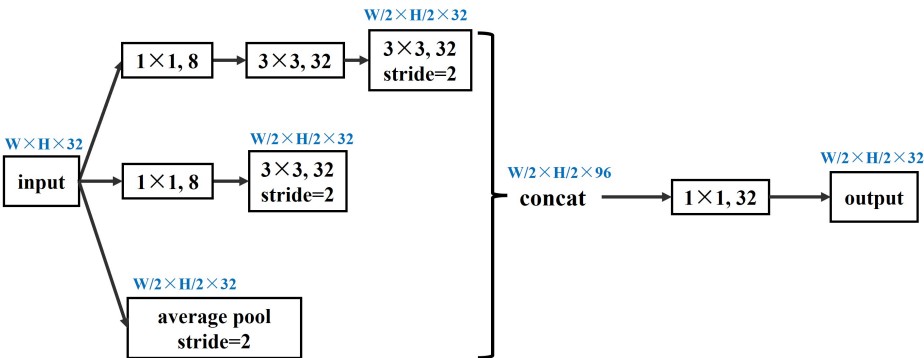

**Figure 2.** The procedure of efficient down-sampling.

### 2.2. Four-Dimensional Cost Volumes Construction

After obtaining multi-scale left and right feature maps through the feature extraction module, it is crucial to fuse them to generate the corresponding matching cost volume. This process allows for the exploration of potential match points within a specified disparity range. A 3D cost volume can be constructed by calculating L1, L2 or a correlation distance between the left feature map and the corresponding right feature map [20,30,31]. In this way, the dimension of the feature channel (C) is compressed and the generated volume has only three dimensions: height (H), width (W) and disparity (D). Using this method can improve the training speed by reducing the number of parameters. However, it reduces the complexity and generalizability of the network, which is not suitable for optical satellite imagery.

We decided to retain the feature channel information and fully exploit the multi-scale features extracted in the previous step. Accordingly, we construct 4D cost volumes independently in four scales, where the dimensions are height (H), width (W), disparity (D) and feature channels (C). The procedure is illustrated in Figure 3. We construct cost

volumes by computing the absolute differences between the features of the left image and the corresponding features of the right image at different disparities. This approach is similar to the result obtained by directly concatenating two vectors [24], but it reduces the number of channels by half, resulting in a more efficient use of computational memory.

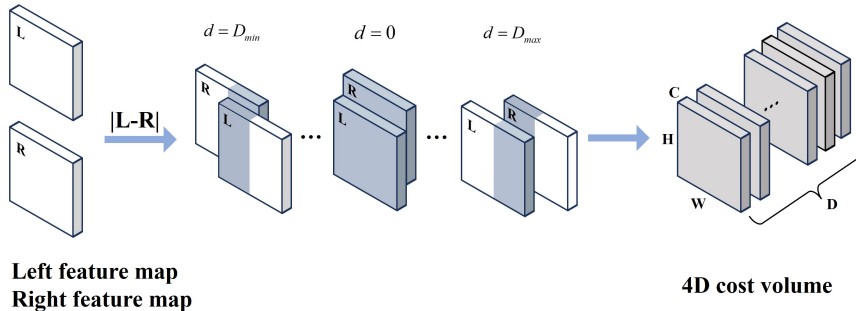

**Figure 3.** The procedure of 4D cost volume construction.

Many networks typically set the disparity range between non-negative values when constructing volumes, but remote sensing datasets may have negative disparity ranges. To account for this and ensure realism, we follow the guidelines outlined in [32] to include the case of negative disparity in the process. It is worth noting that, when the disparity is $-d$, the correspondence between potential matching points $p_l$ and $p_r$ in the left and right feature maps becomes $p_r = p_l + d$. The final size of the cost volumes is $1/2^{i+1}H \times 1/2^{i+1}W1/2^{i+1}(D_{max} - D_{min}) \times 32$, where $D_{min} < 0$.

### 2.3. Multi-Scale Cost Volume Fusion

The 4D feature volumes require 3D convolution to learn regularized aggregation functions over three dimensions: height, width and disparity. We use 3D convolution to construct a stacked hourglass structure integrated with a volume pyramid pool to generate features. The architecture shown in Figure 4 is specifically designed to aggregate multi-scale features from top to bottom.

Starting with the lowest resolution cost volume, a 3D convolution kernel with a stride of 2 is used to down-sample. After capturing smaller-scale features, transposed convolution is used to restore scale and generate the processed cost volume at that resolution. The transposed convolution operation is also used to up-sample to a higher resolution, gradually contributing to the initial cost volume at the same resolution. At the same time, as indicated by the gray arrow in Figure 4, the cost volumes are summed at the same resolution. Each 3D convolution layer is followed by batch normalization (BN) [33] and rectified linear unit (ReLU) activation. When a cost volume addition is encountered, the addition operation is performed first, followed by activation.

We integrate the attention mechanism during the final convolution of the cost volume at each scale, using the SE block [34]. While the original SE block is designed for 3D feature volumes, we extend it to the 4D feature volume. The implementation process is shown in Figure 5. The feature compression process begins with global average pooling, which consolidates the cost volume of $H \times W \times D$ in each feature channel into a single real number. Subsequently, two fully connected layers are employed to generate weights for each feature channel. The parameter $r$ acts as a scaling factor to reduce the number of channels. Different activation functions are applied to increase the non-linearity of the network. Finally, the weights obtained are multiplied by their corresponding channels, completing the recalibration of the original channels and generating the output. This step evaluates the importance of features in different channels.

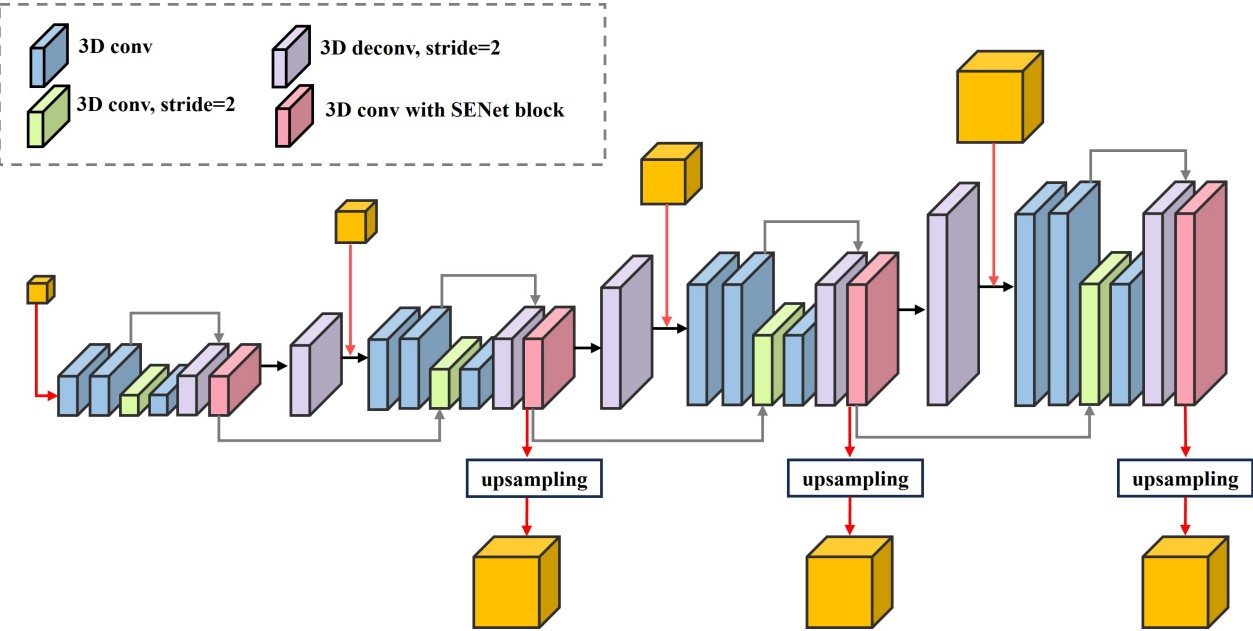

**Figure 4.** Top–down cost aggregation model.

Due to the excessively low resolution at the 1/32 scale, the fused contextual information is limited. Up-sampling to the original resolution during disparity refinement in Section 2.5 results in significant loss of content, which hinders effective restoration of detail. Therefore, we choose not to pass the cost volume at this resolution to subsequent processing steps. Instead, we up-sample the cost volumes at 1/8 and 1/16 resolutions to 1/4 resolution, followed by two convolution operations, resulting in three aggregated cost volumes of equal size.

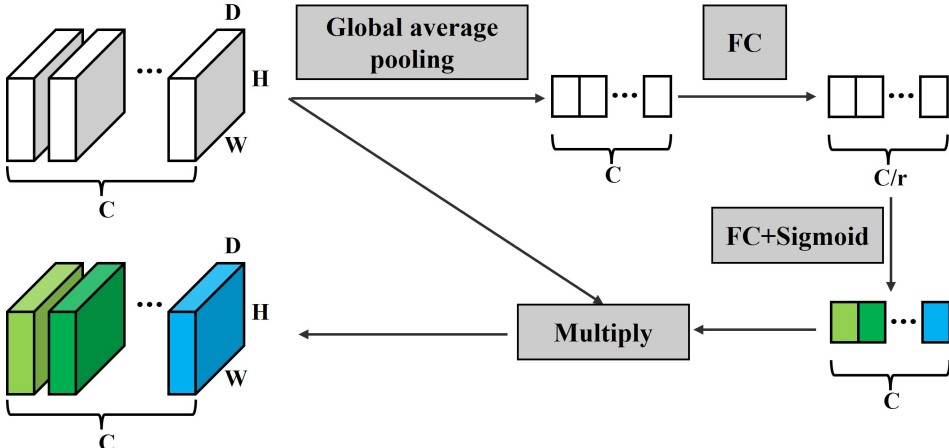

**Figure 5.** The structure of the SE block in the cost aggregation module.

### 2.4. Disparity Regression

After the multi-scale aggregation of the cost volumes, it is essential to compress the feature channels to 1 before proceeding with the disparity regression. We adopt the widely used disparity regression method proposed in [22]. This method uses softmax to normalize the cost values for each disparity $d$, and then uses the normalized probability values to weight the disparities within the disparity range. The final result is a floating-point number that represents the predicted disparity at the subpixel level, thereby improving disparity accuracy. The approach is implemented using Equation (1).

$$\hat{d} = \sum_{d = minD}^{maxD} d \times \sigma(-c_d) \tag{1}$$

where $\hat{d}$ represents the predicted disparity and $\sigma$ denotes softmax, the variable $c_d$ in the cost volume stores the matching cost for a candidate disparity value $d$. A larger matching cost indicates a less favourable match, so it is necessary to take a negative value to obtain smaller probability values through the softmax operation.

*2.5. Edge-Guided Disparity Refinement*

Typically, in most networks, the first step is to estimate low-resolution disparity maps using stereo disparity computation. Interpolation methods are then used to up-sample the maps to full resolution. However, the up-sampling process has inherent limitations that inevitably result in the loss of fine detail. Commonly used linear interpolation methods include nearest neighbor interpolation and bilinear interpolation. These methods use the same kernel during interpolation and do not consider the location of the pixel. In edge regions where there are significant variations in disparity, maintaining clear edges after interpolation becomes a challenge, especially if the disparities do not conform to smoothness assumptions. It is therefore common to introduce additional information into the network to restore the content and texture of the original images. The guided filter [35] is an edge-preserving filter commonly used in traditional methods. By using an input image as a guidance map, it preserves edge details during filtering and finds application in classic tasks such as denoising and detail enhancement. Building upon this, a joint up-sampling guided filter layer is proposed in [36], where the original guided filter is transformed into a convolutional block with learnable parameters. This construction allows for a seamless integration with the CNN, optimizing the whole system through end-to-end training.

The image $O_h$ obtained by the traditional guided filter can be represented by:

$$O_h = A_h \cdot I_h + b_h \tag{2}$$

where $I_h$ represents the guidance map, and $A_h$ and $b_h$ can be calculated from the initial image to be filtered using relevant formulas derived by deduction. The fast end-to-end trainable guided filter replaces traditional formulas with convolutional networks when solving for $A_h$ and $b_h$. This substitution allows it to learn more accurate values. The overall structure is shown in Figure 6. The process starts by applying a mean filter to the input low-resolution guidance map and the low-resolution disparity map. This gives the mean values of the low-resolution guidance map, the mean values of the low-resolution disparity map, the cross-correlation and the auto-correlation from Equation (3).

$$\begin{cases} \mathrm{mean}_{I_l} = f_{\mathrm{mean}}(I_l) \\ \mathrm{mean}_{O_l} = f_{\mathrm{mean}}(O_l) \\ \mathrm{var}_{I_l} = f_{\mathrm{mean}}(I_l \cdot I_l) - \mathrm{mean}_{I_l} \cdot \mathrm{mean}_{I_l} \\ \mathrm{cov}_{I_l O_l} = f_{\mathrm{mean}}(I_l \cdot O_l) - \mathrm{mean}_{I_l} \cdot \mathrm{mean}_{O_l} \end{cases} \tag{3}$$

To reduce computational complexity and increase efficiency, $A_h$ and $b_h$ are learned from the low-resolution images. The convolution module here consists of three cascaded convolutions with a kernel size of 1 and 32 channels. The first two convolutions are followed by BN and ReLU layers. After obtaining $A_l$ and $b_l$, they are further up-sampled to high resolution. Finally, $A_h$, $b_h$ and the input high-resolution guidance map $I_h$ are substituted into Equation (2) and the filtered high-resolution disparity map is obtained by linear model.

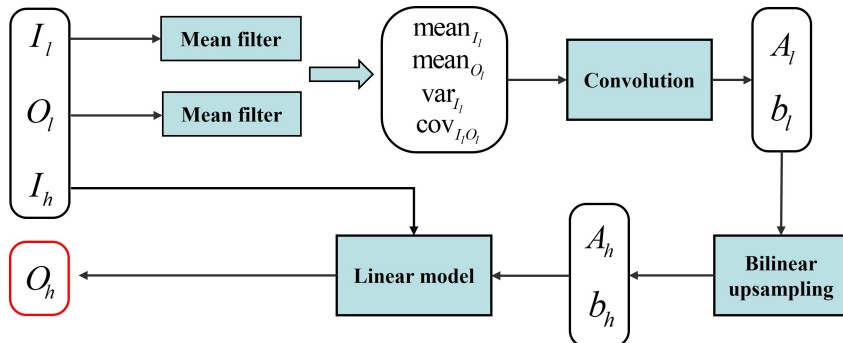

**Figure 6.** Trainable guided filter module.

We extend the fast end-to-end trainable guided filter and integrate it into our model to improve the matching performance, especially at the edges of the terrain. The left image is chosen as the reference image and the necessary features are trained to generate the guidance map, as shown in Figure 7. Given our goal of refining the disparity map during the up-sampling process, the guidance map derived from the trained left image should prioritize edge details with notable disparity variations. The feature extraction module consists of four $3 \times 3$ convolutional layers. Each convolutional layer is followed by a BN layer and a ReLU layer. The resulting high-resolution guidance map $I_h$ is down-sampled to 1/4 of the original resolution. Subsequently, it is separately input, along with the high-resolution guidance map and the low-resolution disparity map, into three guided filters. Finally, three high-resolution disparity maps are generated as output. It is essential to note that, once the disparity map has been restored to its original resolution, the disparity values need to be scaled up accordingly.

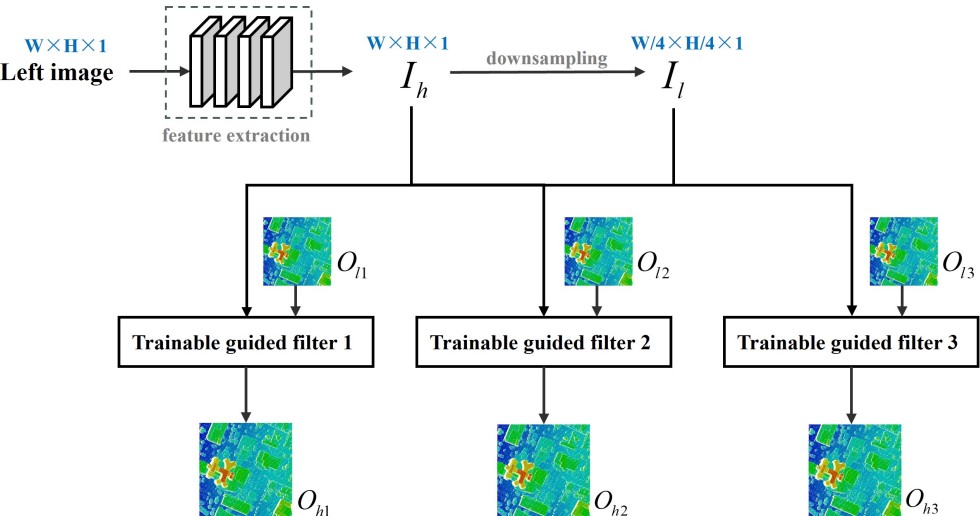

**Figure 7.** The process of disparity refinement.

### 2.6. Loss

We compute different disparity maps at multiple scales, based upon which we calculate the loss function for back-propagation. Due to the aggregation process in the previous step, the high-resolution cost volume incorporates prior information from the low-resolution cost volume, enabling the generation of more refined disparity maps. In addition, the process of up-sampling to the original image size can result in the loss of fine detail. Higher-resolution disparity maps are more likely to produce accurate results than their lower-resolution counterparts. Therefore, we define the total loss as the weighted sum of the losses obtained at each scale:

$$L_s = \omega_1 L_1 + \omega_2 L_2 + \omega_3 L_3 \tag{4}$$

where $L_1$ represents the loss of the disparity map computed from the cost volume at the lowest resolution (1/16), $L_2$ represents the loss of disparity map computed from the cost volume at 1/8 resolution, and correspondingly, $L_3$ represents the loss of the disparity map obtained from the cost volume at the highest resolution (1/4). The variables $\omega_1$, $\omega_2$ and $\omega_3$ denote the weighting coefficients for each loss, and their magnitudes depend on the resolution of the cost volumes. The loss function for the disparity map at a single resolution is defined as:

$$L = \frac{1}{N} \sum_{(x,y)} \text{smooth}_{L_1} \left( \tilde{d}_{(x,y)} - \hat{d}_{(x,y)} \right) \tag{5}$$

where $N$ is the number of valid pixels. When constructing optical image datasets, the ground truth disparity maps often contain some void points. Therefore, when calculating the loss function, it is necessary to exclude these void points and only consider the loss for valid disparities. $\tilde{d}$ represents the ground truth, and $\hat{d}$ represents the predicted value. $smooth_{L_1}$ can be expressed as:

$$smooth_{L_1}(x) = \begin{cases} 0.5x^2 & if \ |x| = 1 \\ |x| - 0.5 & otherwise \end{cases} \tag{6}$$

## 3. Materials

### 3.1. Datasets

There are many publicly available datasets for stereo matching, with KITTI 2012, KITTI 2015 and SceneFlow being widely used. The first two are derived from real-world scenes in a driving environment, including vehicles and numerous street scenes. However, the training sets of these datasets may not be comprehensive. SceneFlow, on the other hand, is a large-scale synthetic dataset that provides dense and accurate ground truth disparity maps. In summary, these datasets all consist of ground scenes that differ significantly from optical satellite data. The optical satellite images have large coverage areas, long baselines and numerous occluded regions, making it difficult to obtain accurate ground truth disparity maps. Therefore, datasets of this type are not abundant. After researching, we adopted the two datasets used in [32] for training and testing. Some information about the two datasets is shown in Table 3.

1.  Urban Semantic 3D (US3D): This dataset was provided in the Data Fusion Contest 2019 and includes more than 4000 pairs of stereo images collected by the WorldView-3 satellite. It covers approximately 100 square kilometers in Jacksonville, Florida, and Omaha, Nebraska, USA, and includes details such as houses, vegetation and roads. For each image block, we only use the provided pair of rectified RGB images together with the corresponding ground truth disparity map. The dimension for both is 1024 × 1024, and the ground sample distance (GSD) is approximately 30 cm. We use 3000 pairs of image data from Jacksonville and Omaha for training, with the remaining pairs reserved for testing.

2.  WHU-Stereo [37]: This dataset is generated from airborne LiDAR point clouds and high-resolution stereo images captured by the Chinese GaoFen-7 satellite. The WHU-Stereo dataset consists of 1700 pairs of images, each with a dimension of 1024 × 1024. The images are 8-bit single-channel panchromatic images. The disparity map is a single-channel image stored in 16-bit float format, covering six regions in China named Kunming, Qichun, Wuhan, Hengyang, Shaoguan and Yingde. The scenes include different types of landscape such as buildings, roads, rivers, farmland and forests. We use over 1000 pairs for training and the remaining pairs for testing. The dataset includes stereo image pairs with relatively long baselines, which poses a significant challenge to the matching process.

**Table 3.** Information about datasets.

| Dataset | Size | Data Segmentation | |
| --- | --- | --- | --- |
| | | Train | Test |
| US3D | $1024 \times 1024$ | 3100 | 1100 |
| WHU-Stereo | $1024 \times 1024$ | 1300 | 410 |

### 3.2. Implementation Details

The network architecture proposed in this study is implemented using PyTorch 1.6.0 and trained end-to-end with the Adam optimizer ($\beta1 = 0.9$, $\beta2 = 0.999$). During data pre-processing, we convert RGB images from US3D to float-type gray-scale images to maintain consistent formatting with WHU-Stereo. Additionally, color normalization is applied to each image. To preserve the integrity of the features in the images, we refrain from cropping the images and input the $1024 \times 1024$ stereo image pairs directly into the network. First, training is performed from scratch for 100 epochs using the US3D dataset. The disparity range is set to $[-96, 96]$, and the initial learning rate is set to 0.001. Then, every 10 epochs, the learning rate is reduced to half. At the end of training, the best weights are selected and evaluated on the test set.

We fine-tune the model on the WHU-Stereo dataset using the pre-trained model from US3D, with a disparity range set to $[-128, 64]$. The learning rate is initially set to a constant value of 0.001 for the first 20 epochs, and then halved every 10 epochs. The training process consists of a total of 120 epochs. Similarly, after completing the training, the best weights are selected and evaluated on the test set. During the training process, the weighting coefficients $\omega_1$, $\omega_2$, and $\omega_3$ in Equation (4) are set to 0.5, 0.7 and 1, respectively. During testing, the output only includes the full-resolution disparity map recovered from the 1/4 resolution because it is the most believable. The network training and testing are conducted on the Windows 10 operating system, using three NVIDIA Tesla V100 GPUs.

## 4. Results and Analysis

### 4.1. Evaluation Metrics

We use the average endpoint error (EPE) and the fraction of erroneous pixels (D1) as metrics to assess the accuracy of the predicted disparity maps. EPE and D1 are very common metrics, and most end-to-end network evaluations use one or both of these metrics [19,23–27,32]. These two metrics are also used in the well-known KITTI leaderboard and SceneFlow leaderboard. If the ground truth disparity map is denoted as $\tilde{d}_k$, the predicted disparity values as $\hat{d}_k$, the set of valid disparities as $S$, and the number of pixels in the set as $N$, then the EPE and D1 are calculated as follows:

$$\text{EPE} = \frac{1}{N} \sum_{k \in S} \left| \tilde{d}_k - \hat{d}_k \right| \tag{7}$$

$$\text{D1} = \frac{1}{N} \sum_{k \in T} T \left[ \left| \tilde{d}_k - \hat{d}_k \right|, t \right] \tag{8}$$

where $t$ is the error threshold. We set it to 4, 2 and 1 according to [27], resulting in D1-4, D1-2 and D1-1, respectively. D1-4 is the percentage of pixels with errors greater than four pixels, which are considered unreliable erroneous pixels. D1-2 and D1-1 represent the percentages of pixels with smaller errors, which are considered tolerable. $T[\cdot]$ is an indicator function used to determine whether pixels meet certain criteria, and it is defined as follows:

$$T[x, y] = \begin{cases} 0, & x < y \\ 1, & x \geq y \end{cases} \tag{9}$$

*4.2. Results*

We compare the results of our proposed EGMS-Net with PSMNet and StereoNet by analyzing the results on two datasets. These networks are classical end-to-end stereo matching networks. After their proposal, they have demonstrated excellent performance on several datasets, and open-source codes have been provided. To control the variables and eliminate the interference of other factors, we employ the same training strategy. This includes using the same parameter initialization, the same learning rate and the same number of training epochs. In addition, the operating systems are kept consistent. Due to the numerous parameters of PSMNet, during training we ensure that the feature dimension layers are moderately reduced based on the size of the GPU without cropping the images, while keeping the network architecture unchanged. This is done to match the feature dimension settings of our network and to preserve the integrity of the training data.

### 4.2.1. US3D

We first predict stereo disparity maps on the test set of the US3D dataset. Table 4 shows the above evaluation metrics and the average time required for processing a single image. Our model saves more memory and reduces the running time by 109 ms compared to the modified PSMNet. Although EGMS-Net is slower than StereoNet, its other metrics are significantly improved. In terms of accuracy, the reduction in network complexity has not impacted the precision of our network. Our network achieves the minimum EPE with only 5.8% of mismatched pixels. D1-2 and D1-1 remain the lowest compared to other networks. In summary, the proposed EGMS-Net significantly improves the matching results compared to StereoNet at the cost of a corresponding increase in training time. Compared to PSMNet, EGMS-Net not only gives better results but also reduces the training time, thus achieving the goal of improving efficiency while maintaining performance.

**Table 4.** The results of different networks on the US3D test set.

| Evaluation Indicator | PSMNet | StereoNet | EGMS-Net |
|:---:|:---:|:---:|:---:|
| EPE (pixel) | 1.582 | 1.772 | 1.515 |
| D1-4 (%) | 6.7 | 8.5 | 5.8 |
| D1-2 (%) | 21.3 | 25.2 | 19.2 |
| D1-1 (%) | 49.6 | 53.3 | 43.6 |
| Time (ms) | 646 | 185 | 537 |

To further illustrate the advantages of our network, Table 5 displays several representative stereo image pairs, with the values of three evaluation metrics (EPE, D1-4 and D1-2) recorded in the table. D1-1's requirements are overly strict, leading to a considerable number of pixels not meeting the criteria; hence, its values are not included in the table. Due to the large disparity range set during training, which may be unfavorable for observing details in a single image, we use different display ranges when presenting disparity maps in Figure 8. This allows for a more intuitive comparison of different methods. The deep blue regions in ground truth disparity maps represent invalid occluded points, and it is necessary to set masks to exclude them when calculating EPE and D1.

The four selected images show urban terrain that is often observed in satellite imagery. The first two images show small patches of vegetation that are commonly found in urban areas. The irregular shapes and undulating surfaces of these vegetation patches result in significant disparity variations. The last two images show different architectural structures. These structures have similar heights, implying consistent disparities, and can be considered as textureless regions. Textureless regions lack distinct pixel intensity changes, which makes feature extraction difficult and has always been a challenge for stereo matching. In addition, sharp edges can cause disparity jumps, making it difficult to compute disparities in edge regions.

**Table 5.** The evaluation results of different networks on representative images from the US3D dataset.

| Figure | EPE (Pixel) | | | D1-4 (%) | | | D1-2 (%) | | |
|--------|--------|-----------|----------|--------|-----------|----------|--------|-----------|----------|
| | PSMNet | StereoNet | EGMS-Net | PSMNet | StereoNet | EGMS-Net | PSMNet | StereoNet | EGMS-Net |
| I | 0.93 | 1.07 | 0.82 | 0.28 | 0.30 | 0.16 | 6.54 | 10.84 | 4.34 |
| II | 0.99 | 1.08 | 0.90 | 3.78 | 3.57 | 2.74 | 12.41 | 12.19 | 9.18 |
| III | 0.83 | 1.32 | 0.75 | 2.51 | 7.71 | 1.30 | 7.78 | 13.38 | 5.90 |
| IV | 1.21 | 1.34 | 1.07 | 0.38 | 3.30 | 0.29 | 12.19 | 11.76 | 9.91 |

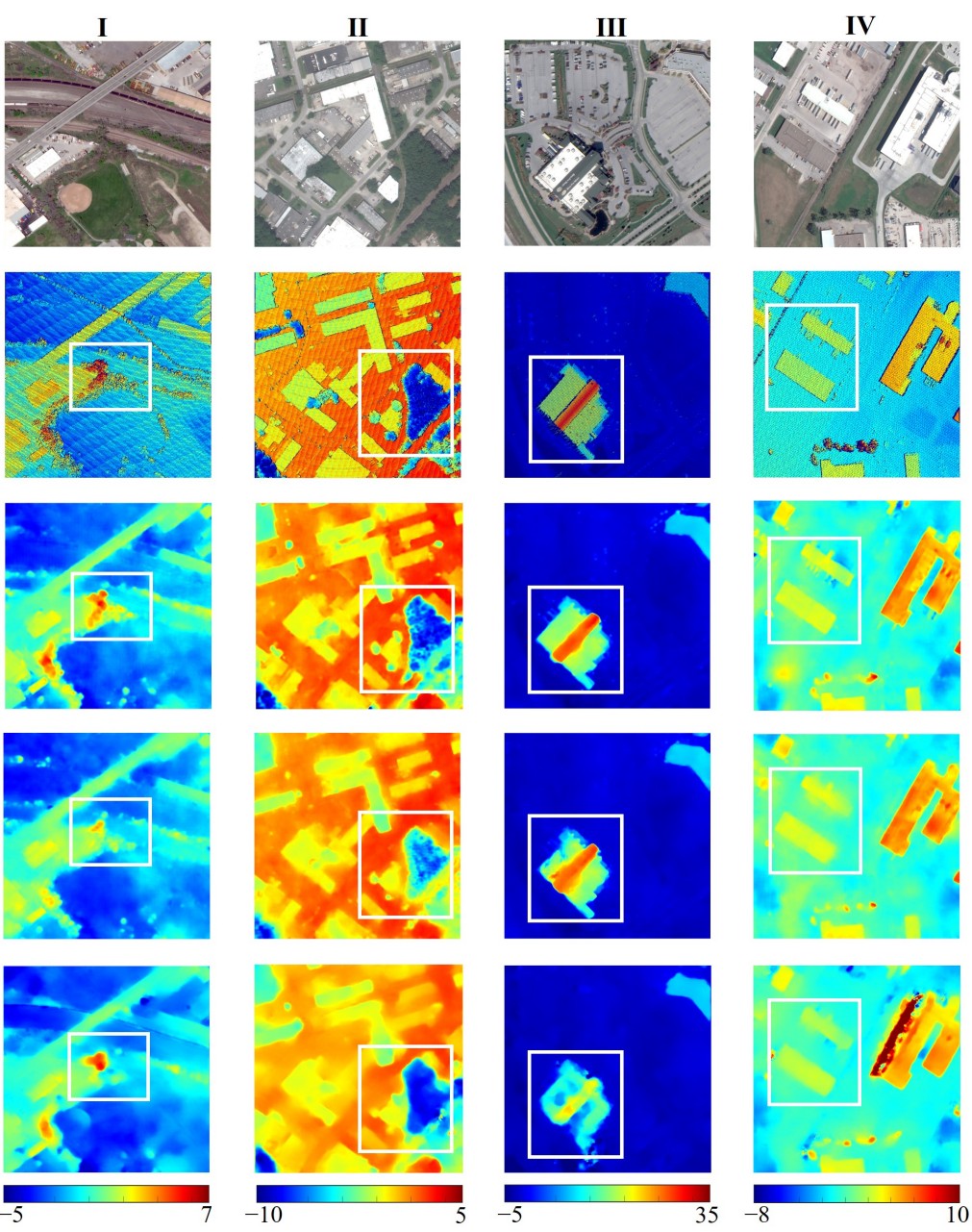

**Figure 8.** The disparity maps generated by different networks on representative images from the US3D dataset. From top to bottom: left image, ground truth, EGMS-Net, PSMNet, StereoNet.

From Figure 8, it can be seen that the disparity maps generated by our network excel at recovering vegetation shapes and providing accurate disparity values. Our network accurately predicts the void points within the vegetation in the ground truth maps. In comparison, PSMNet produces less accurate disparity values and the disparity maps produced by StereoNet show significant shape deviations. In addition, the disparity maps obtained by PSMNet have the problem of blurred edges around buildings. This is due to inaccurate predictions during the up-sampling. In contrast, in the disparity maps obtained by our network, the boundaries where disparities undergo step changes are more regular and correspond well to the ground truth map. The last two disparity maps alleviate the edge expansion problem to some extent. For textureless areas such as building roofs and roads, the disparity changes in the disparity maps generated by EGMS-Net are more uniform. This corresponds to real-world scenarios where disparities show continuity when elevation is constant or changes smoothly. However, StereoNet encounters matching errors in such scenarios and fails to fully recover the shape of the buildings. Looking at the data recorded in Table 5, our network shows better performance in all metrics. In particular, the EPE decreases by 0.14 pixels compared to PSMNet and by 0.57 pixels compared to StereoNet.

Figure 9 illustrates the dense building scenario. It is clear that when the buildings are relatively small, StereoNet disrupts the original arrangement of the buildings and fails to discriminate them effectively. In fact, it fails to identify the buildings in the disparity map VII. This is attributed to the lack of multi-scale feature extraction in the network. The results of PSMNet, which extracts pyramid features, are superior to StereoNet. However, they still show problems such as unclear matching, which leads to blurring of building shapes. Our proposed network achieves the best results. It clearly distinguishes the buildings and predicts more reasonable disparity values.

As a result, our EGMS-Net demonstrates the competence in dealing with challenges related to disparity variations, textureless regions, and dense urban structures. The trainable guided filter in the disparity refinement module is critical in handling edge regions, ensuring precise recovery of edge detail during up-sampling to the original resolution, thereby enhancing the performance of EGMS-Net. In dense building areas, EGMS-Net starts feature learning at the original resolution, ensuring comprehensive capture of high-frequency information. In addition, the top–down cost aggregation module with SE blocks further contributes to its effectiveness. As a result of these network designs, EGMS-Net has advantages in addressing challenging areas of high-resolution optical satellite imagery.

### 4.2.2. WHU-Stereo

We also test the fine-tuned network on the WHU-Stereo dataset. Unlike the US3D dataset, the WHU-Stereo dataset contains more diverse and complex landforms. However, the training data provided by WHU-Stereo are less abundant than that provided by US3D, requiring the model to have a stronger generalization capability. At the same time, the spatial resolution of GF-7 satellite imagery is comparatively lower than that of WorldView-3 satellite imagery. The reduced resolution results in blurred texture detail, which creates challenges for feature extraction. Images of the same size also contain more varied topographic features, which increases the range of disparity variations. Moreover, this dataset includes urban areas with dense and tall buildings, coupled with larger perspectives, which significantly increases the area of occluded regions. Therefore, models trained with WHU-Stereo perform less favorably in the final results compared to models trained with US3D.

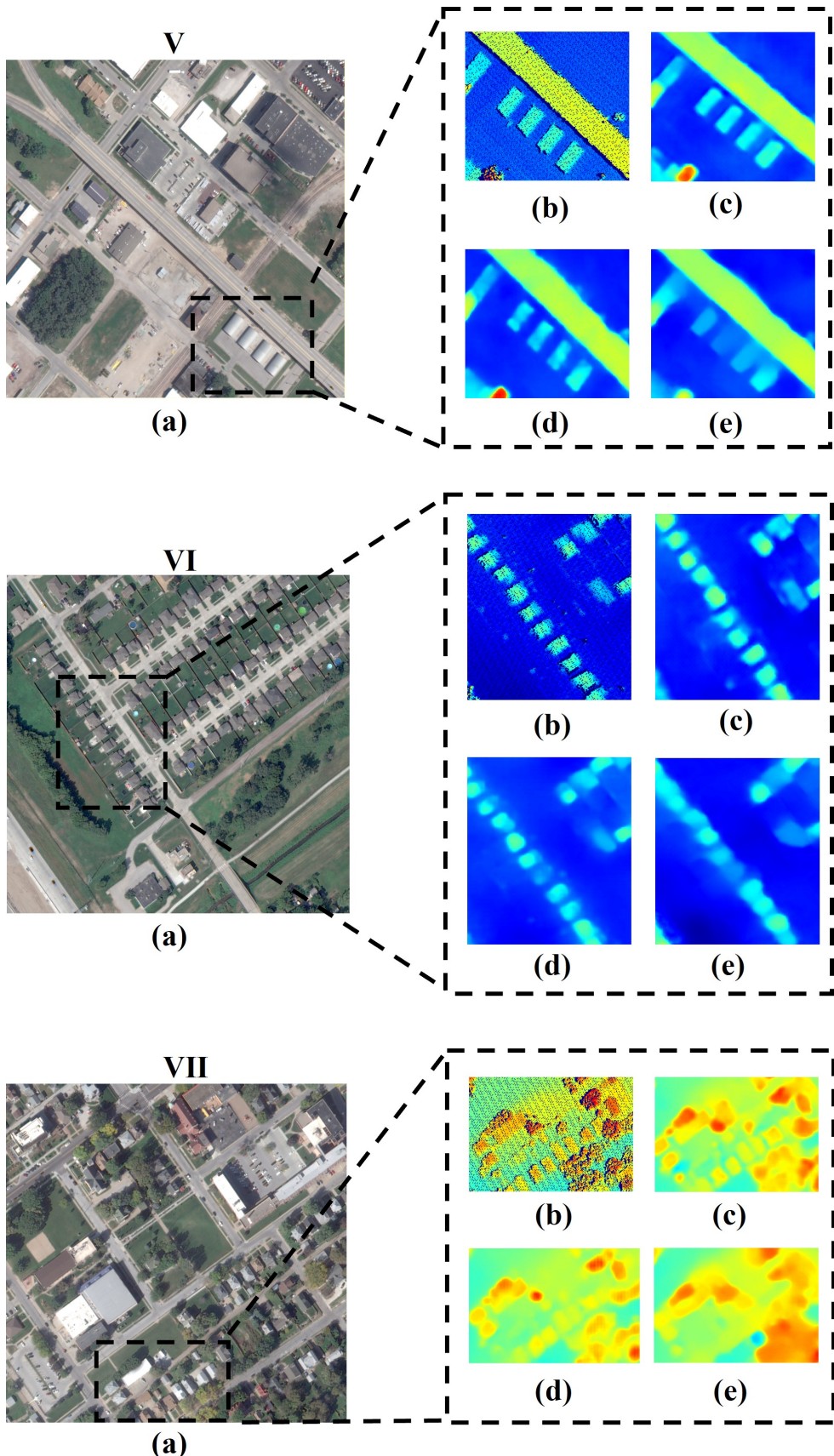

**Figure 9.** Challenging regions within the US3D dataset. V, VI and VII indicate graphic numbers. (**a**) Left image. (**b**) Ground truth. (**c**) EGMS-Net. (**d**) PSMNet. (**e**) StereoNet.

Nevertheless, we still compared the performance of PSMNet, StereoNet and our EGMS-Net on the WHU-Stereo test set, and the results are shown in Table 6. PSMNet and EGMS-Net both show clear advantages over StereoNet. This is because StereoNet uses fewer parameters and a relatively simple model to prioritize real-time performance, making it suitable for applications such as autonomous driving, where maintaining accuracy while improving speed is critical. However, when faced with complex scenes and lower resolution optical images, StereoNet may not be able to cope effectively. PSMNet uses a more complex model and performs well in the final results. Compared to PSMNet, EGMS-Net shows a reduction of 0.377 in EPE, a reduction of 50 ms in running time and the least number of mismatched pixels.

**Table 6.** The results of different networks on the WHU-Stereo test set.

| Evaluation Indicator | PSMNet | StereoNet | EGMS-Net |
|:---:|:---:|:---:|:---:|
| EPE (pixel) | 2.836 | 3.044 | 2.459 |
| D1-4 (%) | 19.1 | 22.5 | 17.6 |
| D1-2 (%) | 43.8 | 53.8 | 42.6 |
| D1-1 (%) | 69.3 | 74.8 | 67.8 |
| Time (ms) | 607 | 184 | 557 |

Figure 10 shows difficult matching regions. The first two satellite images have long baselines, resulting in a disparity range of about 100 within the same scene. This can lead to occlusion in the background areas near tall buildings. Shadows cast by the buildings also cause interference. This can also be seen in the ground truth map where there are large invalid points near tall buildings. These invalid points do not provide any useful information for network training, making it a significant challenge to correctly fill the occluded regions. A common practice is to recover disparity values using nearby information [38], but this can lead to the expansion of building edges. From the results, it can be seen that PSMNet tends to predict the entire occluded area as foreground, resulting in large patches of buildings being connected and degrading the final disparity map results. The results of StereoNet are also poor, with inaccurate predicted disparity values blurring the boundaries of buildings. Our proposed EGMS-Net successfully predicts occlusion as background disparity and preserves the shape of buildings to the maximum extent.

The third image illustrates the low-rise building scenario. There are instances of incomplete building matching in the disparity maps predicted by PSMNet and StereoNet. The fourth image shows plains and mountainous areas with fewer buildings. It can be seen that, in these regions the disparity changes are relatively gradual and sharp edges are rare. Therefore, the disparity maps predicted by all three networks appear satisfactory in these areas. However, upon closer inspection, it can be seen that, in the edge regions of the road, the disparity map generated by EGMS-Net is more closely aligned with the left image, with minimal edge expansion. The data in Table 7 also confirm our observations from the images. Our network achieves the lowest EPE in all four images, demonstrating the superiority of our approach.

Similar to Figure 9, we also select some dense buildings from the WHU-Stereo test set and display them in Figure 11. In the disparity map V, PSMNet produces the worst results. It fails to match many small buildings, resulting in missing information in the disparity map. While StereoNet matches most buildings, it does generate some incorrect disparity values that can be confusing. The sixth image shows that the building boundaries generated by EGMS-Net are the most regular and clear. Edge blur is present in the disparity maps produced by both PSMNet and StereoNet.

**Table 7.** The evaluation results of different networks on representative images from the WHU-Stereo dataset.

| Figure | EPE (Pixel) | | | D1-4 (%) | | | D1-2 (%) | | |
|--------|---------|-----------|---------|---------|-----------|---------|---------|-----------|---------|
| | PSMNet | StereoNet | EGMS-Net | PSMNet | StereoNet | EGMS-Net | PSMNet | StereoNet | EGMS-Net |
| I | 2.66 | 2.83 | 2.38 | 10.91 | 13.96 | 10.09 | 28.35 | 35.55 | 25.93 |
| II | 3.08 | 3.01 | 2.55 | 13.69 | 17.31 | 13.25 | 30.27 | 38.70 | 29.26 |
| III | 1.01 | 1.06 | 0.95 | 3.26 | 4.20 | 2.70 | 9.98 | 12.92 | 10.01 |
| IV | 1.21 | 1.31 | 1.20 | 4.57 | 5.06 | 3.99 | 15.97 | 18.19 | 14.91 |

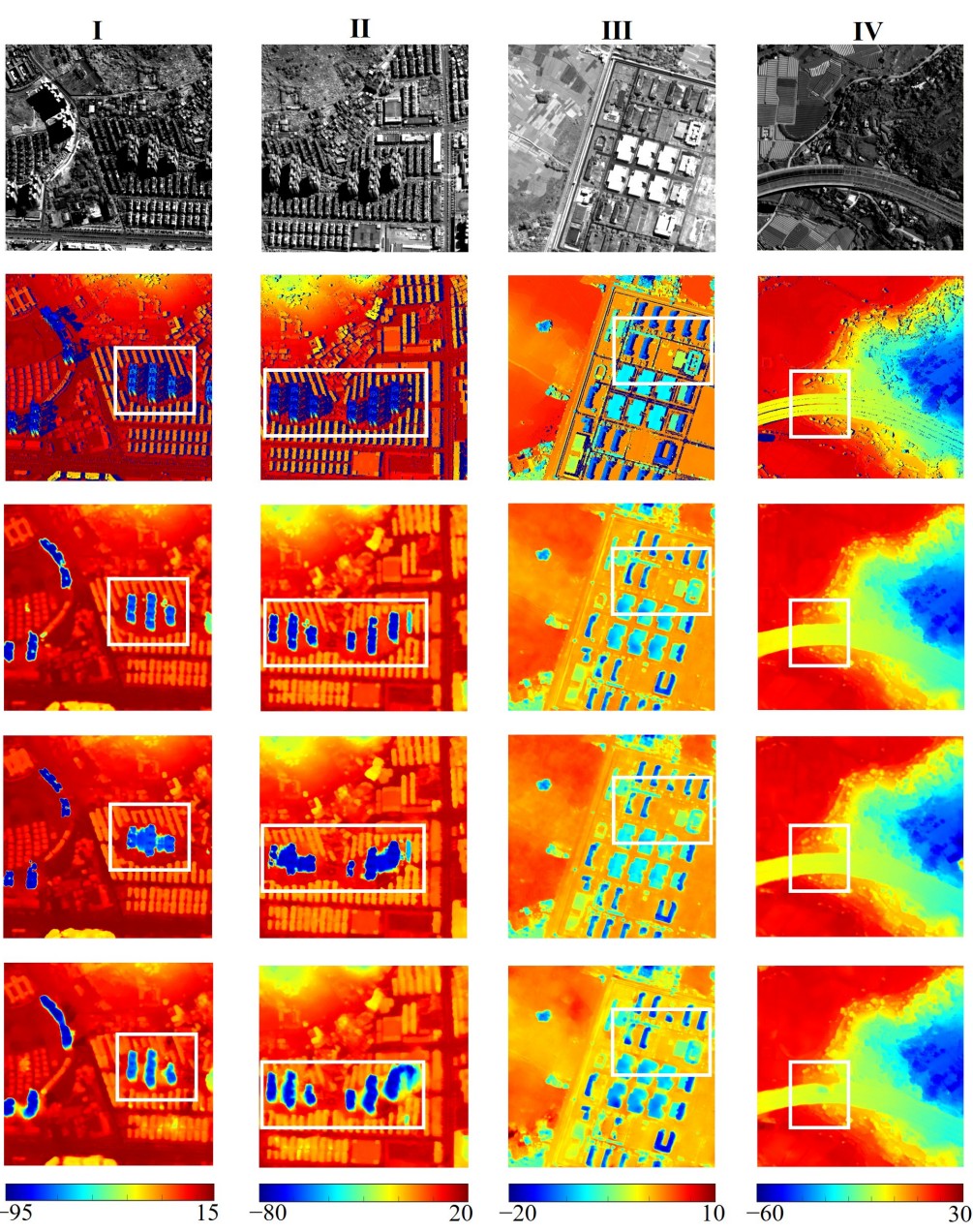

**Figure 10.** The disparity maps generated by different networks on representative images from the WHU-Stereo dataset. From top to bottom: left image, ground truth, EGMS-Net, PSMNet, StereoNet.

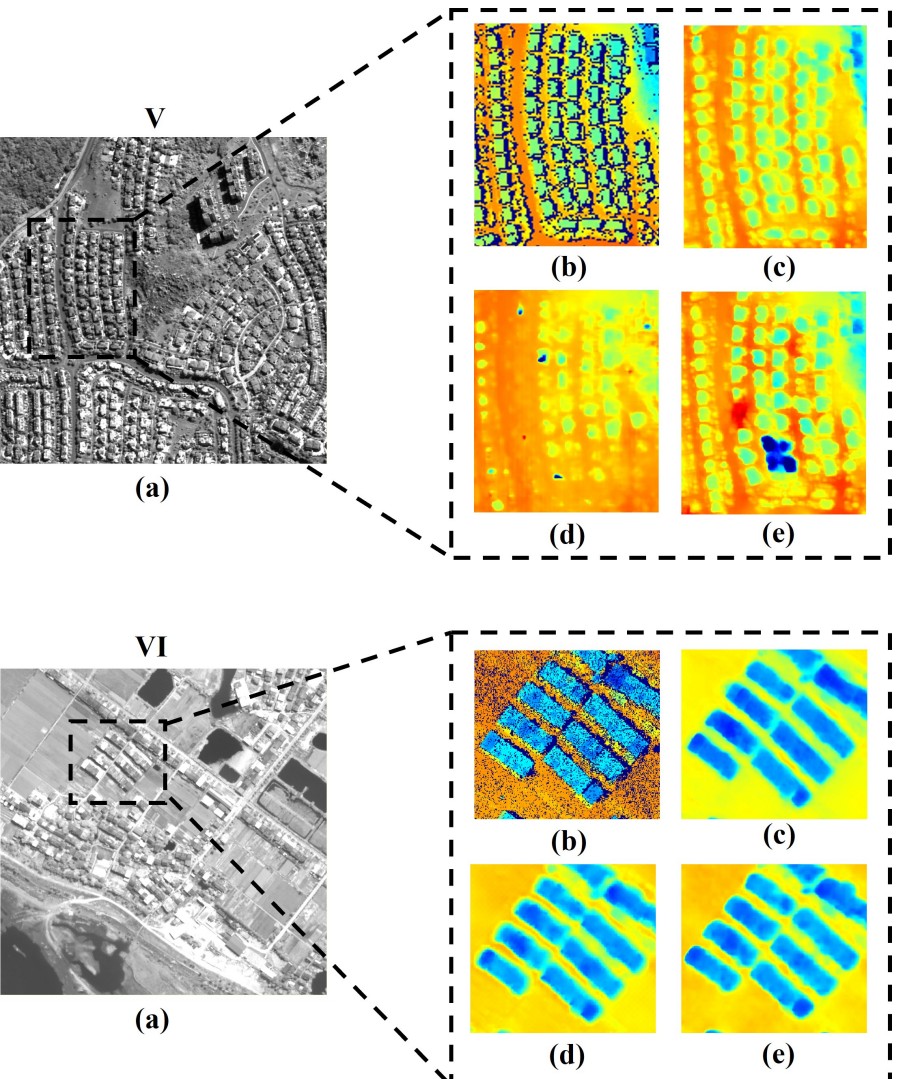

**Figure 11.** Challenging regions within the WHU-Stereo dataset. V and VI indicate graphic numbers. (**a**) Left image. (**b**) Ground truth. (**c**) EGMS-Net. (**d**) PSMNet. (**e**) StereoNet.

## 5. Discussions

### 5.1. Significance Test

To make our experimental results more credible, we perform significance tests on the US3D dataset. Given the long training time, all three models are trained five times with different random seeds, ensuring that the remaining parameters are kept constant. Each of the three methods ultimately produces five results. We test EGMS-Net against PSMNet and StereoNet, respectively. Given the small sample size, we do not expect the difference between the two models to follow a normal distribution, so we use the Mann–Whitney U test in non-parametric form. Table 8 shows the test results of our method with PSMNet and StereoNet, respectively.

**Table 8.** Significance tests of EGMS-Net on the US3D dataset.

| Method | Evaluation Indicator | | | |
|---|---|---|---|---|
| | EPE | D1-4 | D1-2 | D1-1 |
| PSMNet | 0.0317 | 0.0465 | 0.0361 | 0.0317 |
| StereoNet | 0.0079 | 0.0119 | 0.0079 | 0.0079 |

The null hypothesis (H0) states that the metrics of EGMS-Net are not significantly different from those of PSMNet and StereoNet. This implies that the performance of EGMS-Net is similar to that of PSMNet and StereoNet. On the other hand, the alternative hypothesis (H1) states that the metrics of EGMS-Net are significantly different from the metrics of PSMNet and StereoNet. These hypotheses would guide the statistical analysis to determine whether EGMS-Net has superior performance compared to PSMNet and StereoNet based on the defined metrics.

Based on the results presented in Table 8, where the $p$-values for all metrics are less than the significance level of 0.05, the null hypothesis (H0) is rejected. Therefore, the alternative hypothesis (H1) is accepted, indicating that EGMS-Net does indeed differ from PSMNet and StereoNet on these metrics. Judging from the combined data, the metrics of EGMS-Net are smaller than those of the other two networks. This indicates that EGMS-Net achieves better results compared to PSMNet and StereoNet in terms of the defined performance indicators.

### 5.2. Ablation Study

To verify the effectiveness of each component of the network on the results, we perform a detailed ablation study on US3D. We do not change any of the parameter settings during the experiment, including the learning rate and the weights in the loss function. The network is retrained after removing the corresponding components. The results are recorded in Table 9.

**Table 9.** The ablation study by removing each component.

| Method | EPE (Pixel) | D1-4 (%) | D1-2 (%) | D1-1 (%) | Time (ms) |
|---|---|---|---|---|---|
| Full Method | 1.515 | 5.8 | 19.2 | 43.6 | 537 |
| High-resolution | 1.577 | 6.3 | 19.7 | 44.1 | 518 |
| Efficient Down-Sampling | 1.541 | 6.0 | 19.2 | 43.4 | 549 |
| Multi-Scale Aggregation | 1.879 | 9.6 | 26.0 | 56.1 | 396 |
| Trainable guided filter | 1.547 | 6.0 | 19.4 | 43.9 | 549 |

"High-resolution" means that high-resolution feature extraction is not performed at the beginning. Following the other networks, we first reduce the resolution to one-fourth and then perform feature learning. It can be seen that, after this change, the EPE increases by 0.062 pixels and D1-2 and D1-1 also increase. This suggests that learning features at the original size of the input image in the feature extraction module positively contributes to the overall performance of the network.

"Efficient down-sampling" means that we do not use efficient down-sampling, but simply use the normal bilinear down-sampling operation to obtain multi-scale features. As expected, the EPE increases by 0.026 pixels. In addition, although the bilinear down-sampling is simpler, the running time has increased instead. This suggests that this part achieves the goal of improving the efficiency of the network.

"Multi-scale aggregation" means that we do not use multi-scale aggregation, but only acquire the feature maps at 1/32 resolution through efficient down-sampling, and then perform the subsequent aggregation operation. The superiority of multi-scale has been demonstrated in numerous papers. As can be seen from Table 9, the results without multi-scale aggregation are significantly worse, although the running time is reduced.

"Trainable guided filter" means that we do not use the final disparity refinement module, but use bilinear up-sampling to bring the disparity map back to its original size. It can be seen that, although the network becomes simpler when this module is removed, the running time does not decrease, but rather increases. The results of other metrics also become worse. The trainable guided filter is an effective way to increase the matching accuracy.

## 6. Conclusions

This study presents an end-to-end edge-guided multi-scale matching network. The network performs fine feature extraction at the original resolution and constructs feature maps at different scales through efficient down-sampling. Top–down 4D cost volume aggregation is then performed by a feature aggregation module using SE blocks. Finally, a disparity refinement module is used to train the left image to generate the guidance map, while a trainable guided filter ensures accurate edge details when returning to original resolution. In our experiments, EGMS-Net successfully reduces EPE and D1 compared to PSMNet and StereoNet, achieving EPE values of 1.515 pixels and 2.495 pixels on the two test datasets. The primary objective of reducing network running time while maintaining the quality of the disparity map is achieved. EGMS-Net provides notable improvements in challenging regions such as occluded and textureless areas in optical images, particularly in regions of significant disparity variation where it produces sharper edges. Furthermore, the ablation study shows that the four design choices we implemented contribute significantly to the performance of the network.

In future research, we aim to further improve the accuracy of the network, especially in scenarios with larger viewpoints and more occlusions, while at the same time improving its ability to generalize to different datasets. In addition, we recognize the need to address the longer running time of EGMS-Net compared to StereoNet, which warrants further optimization. Furthermore, extending the applicability of the network to lower resolution satellite imagery, such as ZY-3, is an interesting challenge that we plan to investigate.

This study validates the feasibility of using deep learning techniques to construct dense matching networks for optical images. It shows that a favorable balance between processing speed and result accuracy can be achieved, particularly for large format optical images. In addition, the improvements made in EGMS-Net hold promise for application to other computer vision tasks. For example, efficient down-sampling operations can be used to preserve information when generating multi-scale feature maps. If the visualization results need to improve resolution or keep edges sharp, a reference map can be introduced and a trainable guided filter can be added at the end of the network to optimize the final results.

**Author Contributions:** Conceptualization and methodology, Y.L.; Software, Y.L.; validation, Y.L. and H.W.; formal analysis, Y.L.; investigation, Y.L. and H.W.; resources, Y.L. and X.L.; data curation, Y.L. and H.W.; writing—original draft preparation, Y.L.; writing—review and editing, Y.L. and X.L.; visualization, Y.L.; supervision, X.L.; project administration, H.W. and X.L.; funding acquisition, X.L. All authors have read and agreed to the published version of the manuscript.

**Funding:** This research was funded by the LuTan-1 L-Band Spaceborne Bistatic SAR data processing program, grant number E0H2080702.

**Data Availability Statement:** In this study, the remote sensing data were obtained from various sources to support our analyses. We accessed Urban Semantic 3D dataset (US3D) from Data Fusion Contest 2019 (DFC2019) (https://ieee-dataport.org/open-access/data-fusion-contest-2019-dfc2019) (accessed on 15 October 2023), and dataset from WHU-Stereo (https://github.com/Sheng029/WHU-Stereo) (accessed on 21 September 2023). These diverse data sources played a crucial role in our research and provided a comprehensive foundation for our remote sensing investigations.

**Conflicts of Interest:** The authors declare no conflicts of interest.

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
