# Peer review of "End-to-End Edge-Guided Multi-Scale Matching Network for Optical Satellite Stereo Image Pairs"

_remotesensing, doi:10.3390/rs16050882_

Round 1

Reviewer 1 Report

Comments and Suggestions for Authors

The authors developed this study to propose an end-to-end edge-guided multi-scale matching network for optical satellite stereo image pairs. However, I have some concerns.

Include the literature review section according to the following guidelines.

This section must discuss any shortcomings or challenges in prior research. If the previous work is free of challenges and there is no problem, what motivated the authors to conduct this study? According to the guidelines, authors should review the literature review. They should also point out the problems in the previous study, compare the proposed solution, and explain how the proposed solution is best applicable. Additionally, please include a table summarizing related work, weaknesses, and the proposed solution. An accurate guideline can be obtained from the following work, which summarizes previously relevant work in "Table 1".

Most importantly, the authors didn't cite the equations numbers, figures, and tables in the text. I recommend they insert the references of equations, figures, and tables where these are being discussed. Further, when writing abbreviations, must write the full names when these first appear on the paper. Further, the full form of abbreviations must be used in the manuscript.

How are the experimental materials described in Section 3?

What results and analysis are presented in Section 4?

How is the guided filter used in the edge-guided disparity refinement process in the proposed network?

What is the formula used in the proposed network for improving the accuracy of predicted disparity at the subpixel level?

How does the proposed model integrate global information into the network to optimize matching results and maximize computational efficiency?

Can the end-to-end edge-guided multi-scale matching network accurately estimate high-resolution disparity maps from stereo image pairs while also preserving important details and edges? Provide evidence to support your answer.

How might the proposed guided filter-based edge-preserving disparity refinement process be used in other computer vision applications beyond stereo matching, such as image processing or object detection and recognition?

Go for a thorough proofread of the paper to rectify several existing typos and grammatical mistakes to improve the written quality of the paper. If necessary take the help of a native English speaker to improve the language of the paper.

Regarding the conclusion paragraph, Please precisely describe the outcome of the study and justify the statements that are mentioned in the abstract. Further, it must contain additional points and must give a clearer and more discussion about the experimental results. The main novelty and contribution of needs must be summarized and the recommendations based on the obtained results. These results are the hallmark for future extension therefore, please spend some more time writing the conclusion and based on the results suggest new directions.

The reference [15] is neither new nor relevant.

Comments on the Quality of English Language

Language check is required.

Reviewer 2 Report

Comments and Suggestions for Authors

please check the pdf attached, thank you

Comments on the Quality of English Language

minor typos but fine

Reviewer 3 Report

Comments and Suggestions for Authors

The paper introduces an innovative approach called Edge-Guided Multi-Scale Matching Network (EGMS-Net) for obtaining disparity maps in optical satellite stereo image pairs. Dense stereo matching is crucial for generating digital surface models, but the characteristics of optical satellite imagery, such as large occlusions and long baselines, pose challenges to this process. The EGMS-Net utilizes small convolutional filters and residual blocks for effective feature extraction, emphasizing high-frequency signals. Pyramid features are extracted using efficient down-sampling, forming cost volumes. To regularize these volumes, the authors introduced a top-down multi-scale fusion network with an attention mechanism. An inventive use of trainable guided filter layers is employed for disparity refinement, particularly in restoring edge details. The proposed network was trained and evaluated on US3D and WHU-Stereo datasets, showcasing superior performance compared to PSMNet and StereoNet. In challenging scenarios, like textureless regions and dense buildings, EGMS-Net consistently demonstrates satisfactory matching performance. Importantly, the network achieves improved processing speed compared to PSMNet, enhancing efficiency without compromising effectiveness.

Globally, the manuscript is very well written and organized. However, there are some issues that should be addressed.

The introduction section should include (e.g., just before the organization paragraph) the main objectives and contributions of this work.

All figures and tables should be explicitly referenced in the main text; please refer to the attached commented PDF document for specific examples where this is not done.

The parameter L3 in equation 4 should also be explained (as L1 and L2 are).

Section “5. Discussions and Conclusions” presents no discussions; this section only presents the main conclusions.

Please refer to the attached commented PDF document for some other minor issues that should be addressed.

Comments on the Quality of English Language

Please refer to the comments above.

Round 2

Reviewer 1 Report

Comments and Suggestions for Authors

satisfied

Comments on the Quality of English Language

satisfied

Reviewer 2 Report

Comments and Suggestions for Authors

The reviews are fine so the paper can be accepted